# What Will Retirement Pensions Be Like? Analysis of Spanish Future Pensioner Households in Terms of Poverty

**Blanca Urbano [1], Antonio Jurado [2],\* and Beatriz Rosado-Cebrián [3]**

[1] Faculty of Business, Finance and Tourism, University of Extremadura, 10003 Cáceres, Spain; burbanoo@alumnos.unex.es

[2] Department of Economics, School of Technology, University of Extremadura, 10003 Cáceres, Spain

[3] Department of Financial Economy and Accountancy, School of Technology, University of Extremadura, 10003 Cáceres, Spain; brosadot@unex.es

\* Correspondence: ajurado@unex.es

**Abstract:** The Spanish public retirement pension system, the same as that of many European countries, faces two important risks in the long term. On the one hand, the sustainability of the current pay-as-you-go system and, on the other hand, the ability to maintain an acceptable standard of living for the retired population. This paper presents a study on the current situation of the Spanish public retirement pension system and its effect on the future retired population. In recent years, the concern for the long-term sustainability of the system, which is based on pay-as-you-go and defined benefit, has been very present. For this reason, two major reforms were carried out in 2011 and 2013; however, different investigations have indicated the reduction in future retirement pensions as a possible consequence. Regarding this dilemma, this paper aims to study the future poverty risk of the retired population due to the current formulation of the system, by conducting, for this purpose, an analysis of the purchasing power of future pensioners based on the EU-Statistics on Income and Living Conditions (SILC) 2016 of the National Institute of Statistics of Spain. As a result, a future reduction in the replacement rate was observed, affecting the younger population to a greater extent, as well as an increase in poverty in pensioner households using two different scenarios.

**Keywords:** pay-as-you-go; reforms; poverty; retirement pension; European Union Statistics on Income and Living Conditions (EU-SILC)

## 1. Introduction

The Spanish public retirement pension system is based on pay-as-you-go and defined benefit. The fact that it is pay-as-you-go implies that it is based on intergenerational solidarity, so that contributions (the system's income) are used to cover the expenditure on contributory pensions for the same period [1]. In addition, as it is a defined benefit system; at each moment, the percentage that the first retirement pension represents within the regulatory base is defined, which is obtained according to the calculation rules in force at the time of retirement. These rules are applied to various parameters, work history data (contribution years, contribution bases, and the age of accessing to a retirement pension) [2].

In addition, the system is contributory, and therefore "there is a correspondence between the contributions made during working life and the pensions received during the retirement period, so the more you contribute, the higher the pension" [3].

As in the case of Spain, there are many countries in the European Union that have a system of these characteristics, as it is one of the most widespread modalities [4]. However, a large number of countries, including Spain, have had to undertake different reform processes of their public pension systems, mainly due to the progressive ageing of the population, as well as the economic–financial crisis that began at the end of 2007, which highlighted the shortcomings of these systems [1].

As a result, the Spanish Government introduced two major reforms in the public retirement pension system in 2011 and 2013, through Law 27/2011, of 1 August, on Updating, Adaptation and Modernization of the Social Security system, and Law 23/2013, of 23 December, regulating the Sustainability Factor and the Revaluation Index of the Social Security Pension System, in which measures such as pension revaluation based on the Pension Revaluation Index (PRI) is currently suspended.

The aim of these measures was to ensure the long-term sustainability of the pension system. However, OECD [5] points out, in its report "Pensions Outlook 2018", although the financial sustainability of pension systems increased after their redesign, it is time to focus on the adequacy of the retirement pensions they provide. In this regard, taking into account that the weight of the reforms falls to a greater extent on the current contributing population, different research papers ([6–9], including others) warn about the possible negative effects that these measures may have on the purchasing power of future pensioners, causing a reduction in the average pension to be received, and running the risk of generating pockets of poverty among the retired population.

For all the above, this paper aims to study the effect of the new formulation of the system on the purchasing power of future pensioners, proposing what will then happen with pensioner households, measuring its effect in terms of poverty. Thus, the proposed study allows us to know not only the purchasing power of future pensioners at an individual level, but also its impact in terms of relative poverty in the household as a whole.

To this end, first, a descriptive methodology was used, by reviewing the literature, as well as its study and adaptation. An analytical methodology was then used, with the aim of knowing the effect of the reforms on future pensions, as well as their impact on Spanish households. To carry this out, the European Union Statistics on Income and Living Conditions (EU-SILC) 2016 published by the National Institute of Statistics for Spain [10] were used, studying the income level of its members by age groups, projecting their future pensions and, finally, transferring these projections to their respective homes, obtaining in this way, the effect of the reforms, both at the "pensioner" and "household" level.

The work is structured in five sections. After this introduction, the main modifications introduced in the Spanish pension system after the 2011 and 2013 reforms are presented, as well as their effects on the retired population. In Section 3, the methodology followed with the database used to carry out this study, the EU-SILC 2016, is presented. In Section 4, the data are analysed, and the results obtained are presented and interpreted. Finally, the conclusions derived from this study are discussed.

## 2. The Spanish Public Retirement Pension System: Main Changes and Effects

Considering the projections, and following the guidelines of the "Pacto de Toledo" (Toledo Pact) Evaluation and Reform Report (Parliamentary Commission that was created in 1995 with the support of all political parties with the aim of discussing, analysing and proposing measures aimed at ensuring the sustainability of the Spanish pension system), and approved at the plenary session of the Congress of Deputies on 25 January 2011, the Spanish Government introduced two major reforms in the public pension system, through Law 27/2011, of 1 August, on Updating, Adaptation and Modernization of the Social Security system, and Law 23/2013, of 23 December, regulating the Sustainability Factor and the Revaluation Index of the Social Security Pension System.

As pointed out [11], these reforms have been classified as the broadest and most important carried out to date, their main objective being to ensure the viability of the system in the long term. To that end, the pay-as-you-go system is maintained, with the aim of increasing the correlation between what is actually contributed and the benefit finally received. In addition, a transitional period is established for its gradual application, so that, with the exception of the Revaluation Index, these measures do not affect current pensioners.

In Table 1, main changes are highlighted:

**Table 1.** Main changes in Spanish public retirement pension recent reforms.

| |
|---|
| Increase in the legal retirement age to 67 |
| Increase in the period to be considered in the calculation of the pension regulatory base, from 15 to 25 years |
| Modification of the percentages applicable to the regulatory base |
| Integration of the contribution gaps in the calculation of the regulatory base |
| Establishment of two modes of access to early retirement, differentiating between voluntary or non-voluntary access. These measures have been toughened through Royal Decree-Law 5/2013, of 15 March, on measures to favour the continuity of working life for older workers and promote active ageing. In general, the possibilities of retiring early are: <br><br>(a) At an age four years lower than the applicable legal age (67 or 65, depending on the case) and with at least 33 years of contributions provided that the worker is terminated for reasons not attributable to him/her (involuntary early retirement). <br>(b) At an age two years less than the applicable legal age (67 or 65, depending on the case) and with at least 35 years of contributions, in the case of voluntary retirement. |
| Sustainability Factor (SF): as detailed in Law 23/2013, of 23 December, this instrument makes it possible to link the amount of public retirement pensions with pensioners' life expectancy automatically. Although Law 23/2013 indicates the application of the SF as of 1 January 2019, the reform of the Spanish pension system carried out in 2018 finally delayed its application, setting the deadline for its application for 1 January, 2023 [12] |
| Pension Revaluation Index (PRI): like the SF, the PRI is regulated by Law 23/2013 of 23 December, and has been used since 1 January 2014 to update all pensions, thus replacing the Consumer Price Index (CPI), applied since 1997. It is important to highlight the link of the annual increase in pensions to the financial situation of the system, through the parameters shown below |

Source: own elaboration.

Due to the reforms and as indicated in [8], most of the measures have focused on modifying the system parameters, fundamentally modifying the parameters that directly influence the calculation of the initial retirement pension: age (delaying the legal retirement age), regulatory base (extending the period for calculating the contribution bases), and the percentages applicable to the contribution bases (reduced for the same contribution level), and increasing the number of years required to reach 100%. Furthermore, as stated by [13], most of the measures directly affect the group of contributors, without affecting current pensioners, since, due to their age or disability, the public pension constitutes their main or only source of income.

Similarly, studies such as those by [14] and by [6] indicate that the equity of the system does not improve, in addition to causing a reduction in the average pension to be received as the retirement age or years of contribution are increased, with the risk of generating pockets of poverty among the retired population.

Along the same lines, [4] consider that maintaining the current scenario, and once the Sustainability Factor is applied, the system could achieve a balance as of the second half of the 2020s.

Regarding this last issue, different investigations, such as [7–9,14–17], warn about a possible loss of purchasing power of many cohorts of pensioners as a result of the application of the new automatic adjustment system. Thus, although a favourable result for the financial equilibrium of the system is achieved, it also implies the "quasi"-freezing of pensions for several decades, and as a consequence, their purchasing power depends largely on the evolution of the inflation rate.

Thus, the 2012 and 2015 European Union reports indicate the progress made regarding the sustainability of public pension systems; however, the results regarding their adequacy are not so positive [9].

And the fact is that replacing Consumer Price Index (CPI) by PRI to update pensions has, for the first time, directly and immediately affected current pensioners, arousing significant discontent.

As a consequence, in 2018 the Spanish Government, through the General State Budgets, established an additional revaluation of 1.35% for all contributory pensions for this period, delaying at the same time, the deadline for applying the Sustainability Factor (SF) until 1 January 2023 [12]. In order to maintain pensioners' purchasing power, these measures

were maintained in 2019, establishing the revaluation of pensions for this year according to the CPI, as shown in Royal Decree-Law 28/2018, of 28 December, for the revaluation of public pensions and other urgent measures in social, labour and employment matters. In 2020, the revaluation of contributory pensions for the period resulted in an increase of 0.9% in general terms, regulated by Royal Decree-Law 1/2020, of 14 January, which establishes the revaluation and maintenance of the public pensions and benefits of the Social Security system.

In this context, in 2020 the recommendations of the Toledo Pact aimed at a new reform of the Spanish pension system were finally approved, maintaining the current pay-as-you-go system. However, authors such as [18] assure that the change towards a PAYG system of individual notional accounts (defined contribution) would be the best option to ensure the sustainability and equity of the pension system.

Thus, for all the aforementioned, the main objective of this work is to analyse the possible loss of the purchasing power of the future retired population and its effect on the relative poverty rates in this vulnerable group of the population, taking into account the revaluation of pensions based on the PRI of Law 23/2013, with a detailed study in the following sections.

Table 2 shows the main characteristics of pension systems in European countries. It can be seen that most systems are pay-as-you-go and defined-benefit systems with similarities in relation to retirement age, the number of years of contributions and the revaluation of pensions. Thus, although we are aware that our work focuses on the Spanish pension system, our analysis can be extended to most European pension systems.

**Table 2.** Characteristics of pension systems in Europe.

| | System | Retirement Age | Years of Contributions Required | Years to Receive the Pension | Pensions Index |
|---|---|---|---|---|---|
| Germany | Points System | 65 years-old | 45 years | Working life | Wages and Sustainability Factor |
| Denmark | Defined Benefit | 66 years-old | 40 years | | Wages |
| Finland | Defined Benefit | 63–65 years-old | | Working life | Prices and Wages |
| France | Defined Benefit | 62 years-old | 43 years | Best 25 years of contributions | Prices |
| Greece | Defined Benefit | 65–67 years-old | 40 years | Last 25 years of contributions | Prices and GDP |
| Netherlands | Defined Benefit | 65–67 years-old | | | Wages |
| Hungary | Defined Benefit | 65 years-old | | Working life | Pricess and Wages |
| Italy | Notional Accounts | 67 years-old | 42.1 years: M 41.1 years: F | Working life | Prices |
| Portugal | Defined Benefit | 65 years-old | 40 years | Working life | Prices and GDP |
| Sweden | Notional Accounts | 61–67 years-old | 40 years | Working life | Wages |
| United Kingdom | Defined Benefit | 65 years-old | 35 years | Working life | Prices, Wages and GDP |
| Spain | Defined Benefit | 65 years-old 67 years-old (2027) | 38 years and 6 months 37 years (2027) | Last 25 years of contributions | Prices |

Source: own elaboration based on OECD (2019).

## 3. Methodology

The database selected for the study was the EU-SILC 2016 for Spain [10]. The choice of this base as opposed to others is mainly based on the fact that the Survey provides data, not only at the individual level, but also at the household level. Thus, once the sample has been prepared, it is possible to estimate the contributory retirement pensions to which future pensioners will be entitled, as well as the study of their sufficiency in the entire household.

The database selected for the study was the EU-SILC 2016 for Spain [12].

As shown by the National Statistics Institute (INE) [19], the sample consists of four files, of which the following were used in this study:

&#9675;    File P: A file of detailed data of adults, used in order to understand the average annual income obtained by each of the individuals.

&#9675;    File Q: a detailed data file of adults, used in order to meet the average annual income obtained by each of the individuals.

&#9675;    File H: A file of detailed data of households, in used order to understand the situation in terms of poverty of each of the households to which the individuals belong.

Thus, before proceeding with the analysis of the sample, the data had to be adequately prepared. In Table 3 the different actions are shown.

**Table 3.** Preliminary steps on the EU-SILC sample.

| |
|---|
| Checking that all the individuals in the sample are representative based on the cross-sectional Weighting Factor. |
| Identification of the individuals in each of the households. |
| Source of income as the main filter: PL031 variable (situation in relation to the activity defined by the person concerned). |
| Based on the previous filter, only individuals who are in a contributory situation in the Social Security system (that is, employees or self-employed workers, as well as those declared as unemployed, who will be object of analysis later) are retained. |
| To remain in the sample, they must have at least one of the following income variables: monetary or non-monetary gross income, or profits or losses in the case of the self-employed |
| As for the members declared "unemployed", the benefits received were only taken into account if they are of contributory origin. To differentiate whether their origin is welfare or contributory, the amount of the welfare unemployment benefit for the year 2015 was taken as a reference; that is, 80% of the current Public Indicator of Multiple Effects Income (EUR 532.51 per month). Thus, if the declared amount is higher than EUR 426 per month, the benefit is considered contributory and is incorporated into the analysis. |
| Those individuals receiving unemployment benefit, but not income from work were not part of the study, due to the impossibility of estimating their future income accurately, and, therefore, their first retirement pension. |
| Calculation of the average equivalent income (household disposable income/OECD-modified scale consumption unit). |
| Households whose average equivalent income turned out to be zero or negative were excluded from the sample. |
| Finally, the poverty rates were calculated as 60% of the weighted median of the equivalent income (the official criterion applied by the European Union). |

Source: own elaboration.

## 4. Analysis and Results

After preparing the data and constructing the sample, the analysis was carried out. In the first place, at income level, "workers" and "unemployed" situations were distinguished between. Once the average income was obtained, future pensions were projected by age group. Finally, with the data obtained in the previous sections, the effect of the reforms on the poverty of pensioner households was analysed.

### 4.1. Average Annual Income

First, we calculated the average annual income for each of those aged between 18 and 66, considering in our analysis the best scenario with stable careers, distinguishing between workers and unemployed, and considering different variables as Table 4 shows:

**Table 4.** Variables used to calculate average annual income.

| Gross monetary or quasi-monetary income of the employee in the year prior to the survey |
| --- |
| Non-monetary gross income of the employee in the year prior to the survey |
| Social contributions paid by the employer in the year prior to the survey |
| Gross monetary benefits or losses of the self-employed in the year prior to the survey |
| Gross contributory unemployment benefits in the year prior to the survey |

Source: own elaboration.

For each age analysed (18–66 years) and from the average incomes obtained from the Survey, the rest were obtained up to retirement age based on the projections on wage growth and the CPI of the European Commission (2015).

As shown in Figure 1, the average income follows an increasing trend in both situations, so the lowest wages are found in the first working years, increasing gradually, and the highest wages are found in the ages closest to retirement.

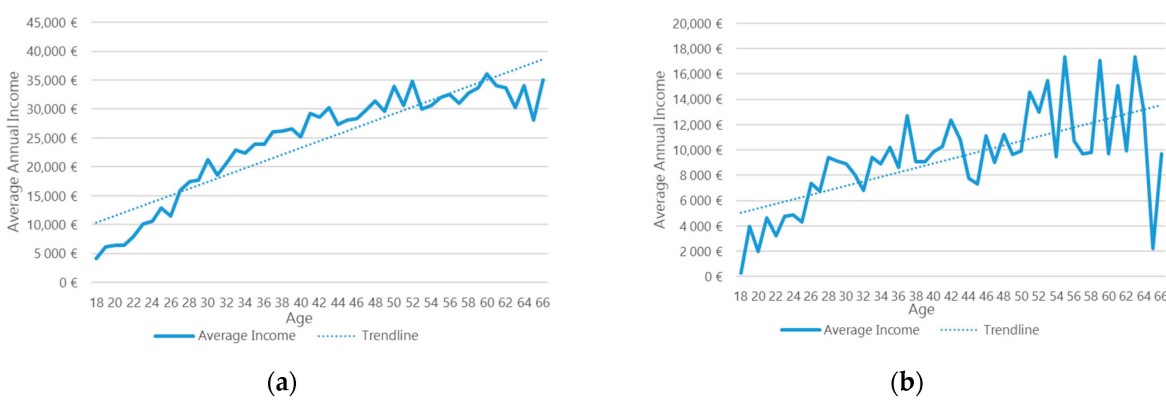

(**a**)　　　　　　　　　　　　　　　　　　　　(**b**)

**Figure 1.** Average income according to age of workers (**a**) and unemployed (**b**). Own elaboration based on EU-Statistics on Income and Living Conditions (SILC) (2016).

However, and according to the data obtained from the Survey, we can see how the average income falls between 60 and 66 years of age, both in the case of the employed and unemployed, which can cause a decrease in the first pension that they will receive. Furthermore, there is a big difference between the incomes in "workers" and "unemployed" situations, which affect their respective contribution bases and, therefore, will be reflected in their future pension.

### 4.2. Average Annual Pension

Once the average income is known, and with it, its respective contribution bases for the different age groups (from 18 to 66), we then proceeded to calculate what the first pension would be for each case, taking into account the 2011 and 2013 reforms, collected in Section 1 of this paper. For this purpose, the working hypotheses detailed in Table 5 were taken into account for the construction of the base scenario.

Thus, taking into account these technical bases, the results obtained in relation to the first retirement pension are shown below. As in the previous case, the study is shown separating the "workers" and "unemployed" situation.

In the case of "workers", it was observed that the first pension obtained follows a slightly decreasing trend as age increases; however, the replacement or substitution rate (relationship between the first pension and the last salary) follows an increasing trend as age increases, as shown in Figure 2. That is, the future Replacement Rate is going to decrease.

**Table 5.** Working hypotheses for base scenario construction.

| |
|---|
| To calculate the amount of the initial pension, the rules established in Law 27/2011 were followed, without taking into account the transitional period. Retirement was considered at the legal age of 67; however, those individuals who reach 38.5 years of effective contribution between 65 and 67 years of age accessed retirement, receiving 100% of their regulatory base. |
| To calculate the year of entry into the labour market, the age of entering was considered to be 18. Similarly, it was assumed that they will receive their first pension when they are 67 years old. If periods without contributions appear in any of the 25 years taken for calculating the regulatory base, such gaps were integrated by following the rules of Law 27/2011. |
| Once the initial pension was calculated, it was multiplied by the Sustainability Factor obtained from Law 23/2013 and based on the most up-to-date Social Security mortality tables, as established by the Law. |
| The consumer Price Index (CPI), based on the data provided by the National Statistics Institute (INE) (2018), estimating its value since 2018 according to the projections of the European Commission (2015) [20], established at 2% per year was used. In the same way, the income was projected based on the Salary Revaluation Index, established at 3% by the European Commission. |
| Limits established for the 2015 contribution bases used were those included in the Official State Bulletin (BOE) [21]. |
| The limit to the maximum and minimum pension used was that included in the BOE for 2015 [22] and increased according to the expected CPI. In addition, the estimated pensions were limited by the maximum and minimum amounts in force at all times. |
| It should be taken into account that individuals do not know their future retirement pension, but they can make an approximate estimate taking into account three known variables: years of contributions, contribution bases and retirement age. |

Source: own elaboration.

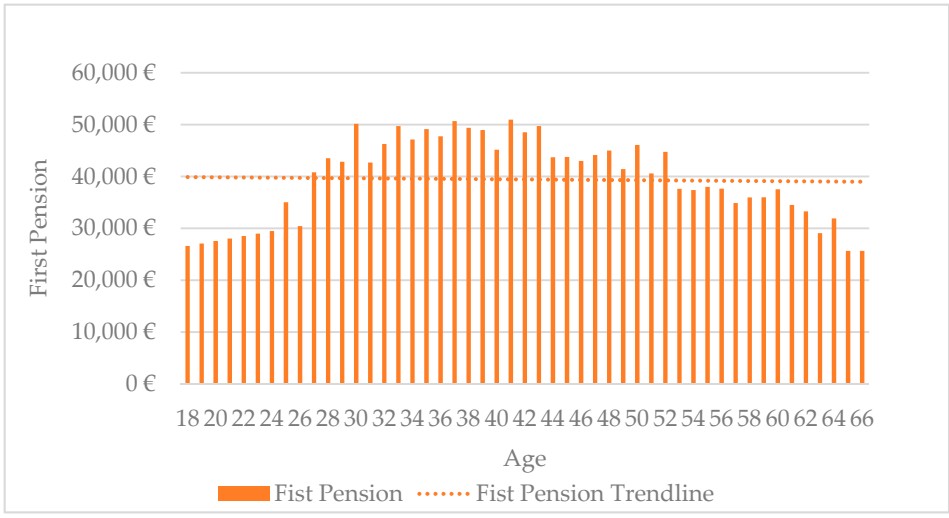

**Figure 2.** First Pension for workers. Own elaboration based on EU-SILC (2016).

This decreasing trend of the future Replacement Rate is similar to that shown for the pension systems of OECD countries. However, while Spain is in the group of countries with the highest replacement rate, the average for the OECD and G20 countries stands at 59%, and ranges from about 30% in Lithuania, Mexico and the United Kingdom, to 90% or more in Austria, Italy, Luxembourg, Portugal and Turkey, at retirement age [23].

Figure 2 shows the first average retirement pension for workers obtained according to the age of the retired population, while in Figure 3 we can observe the trend of the replacement rate for workers, understood as the relationship between the last salary and the first pension that retirees will receive by age. Therefore, the part of the sample that is the furthest from the retirement age would receive at that moment a slightly higher pension than those who are at a closer age to retirement at the time of the survey. However,

when relating this pension to the last salary, the result is the other way around, so although this pension is higher, if compared to their last salary, future pensioners further away in time would receive pensions with a greater difference than those future pensioners that are closer to retiring. Thus, while the replacement rate found for the 60–66 age group is 87–89%, for an average segment of the sample, 41–46 years old, it is reduced to 83%, and for the youngest group, 23–27 years old, 81%, which would imply an additional reduction of 8% in their purchasing power at the time of retirement (Figure 3).

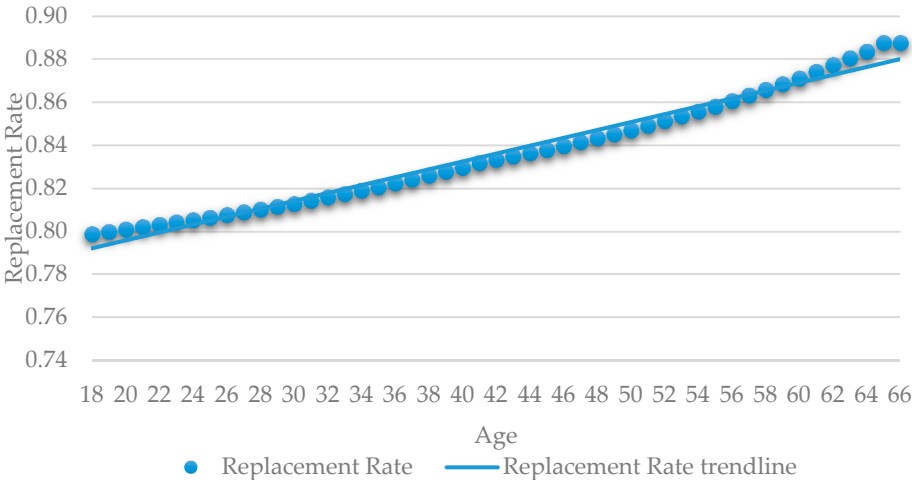

**Figure 3.** Workers´ Replacement Rate. Own elaboration based on EU-SILC (2016).

In the same way, and regarding the youngest group of the sample (between 18 and 22 years old), given that the average income obtained for them has been the lowest, the results show that at the time of their retirement, they would receive the minimum pension estimated for their respective years (Table 6).

**Table 6.** First pension 18–22 years old.

| Age | Entry Year | Retirement Year | Pension EUR/Year |
| --- | --- | --- | --- |
| 18 | 2016 | 2065 | 29,501.42 |
| 19 | 2015 | 2064 | 28,922.96 |
| 20 | 2014 | 2063 | 28,355.84 |
| 21 | 2013 | 2062 | 27,799.85 |
| 22 | 2012 | 2061 | 27,254.75 |

Source: own elaboration.

Regarding the results obtained in relation to the sample corresponding to "Unemployed", as shown in Figures 4 and 5, the amount of the first pension also follows an increasing trend. Thus, in the proposed scenario, the fact that the poverty rate falls can be striking. However, it must be recalled that it is a measurement of relative poverty with respect to a median so, as income decreases, the median decreases, and as a consequence, fewer households can be found below it. For example, this event occurred at the beginning of the crisis in Spain in 2008, the year in which there was a decrease in poverty rates due to the fact that the average monthly wage was reduced and, therefore, the median; however, pensions were maintained, which caused this decline. However, the replacement rate, contrary to the previous case, seems to follow a slightly increasing trend in the future, seeing as the replacement rate trend decreases as age increases in Figure 5. This is due to the projections made, since it has been considered that they will continue with these benefits until the end of their working life, as the probabilities of transition between employment and unemployment were not introduced. As a consequence, the incomes obtained in comparison with the "workers" sample are considerably lower, obtaining in all cases amounts of the first pension equal to or less than the minimum pension calculated for

each period, thus producing an increase in the replacement rate, especially for the youngest ages (18–24 years old), as shown in Figure 5. In this way, we can observe how the pension in the case of the unemployed has been limited by the minimum pension projected for the age, producing in some cases an increase in the replacement rate.

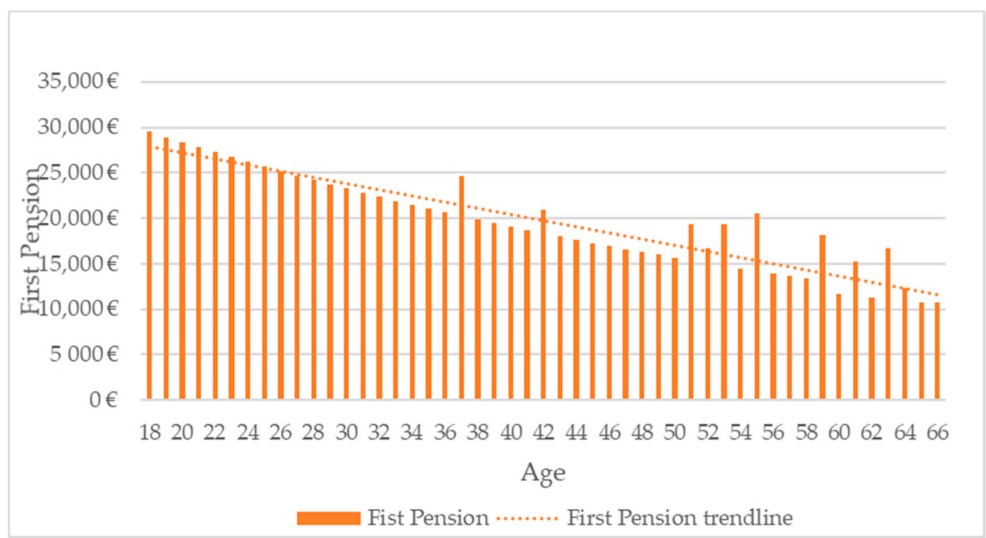

**Figure 4.** First pension for the unemployed. Own elaboration based on EU-SILC (2016).

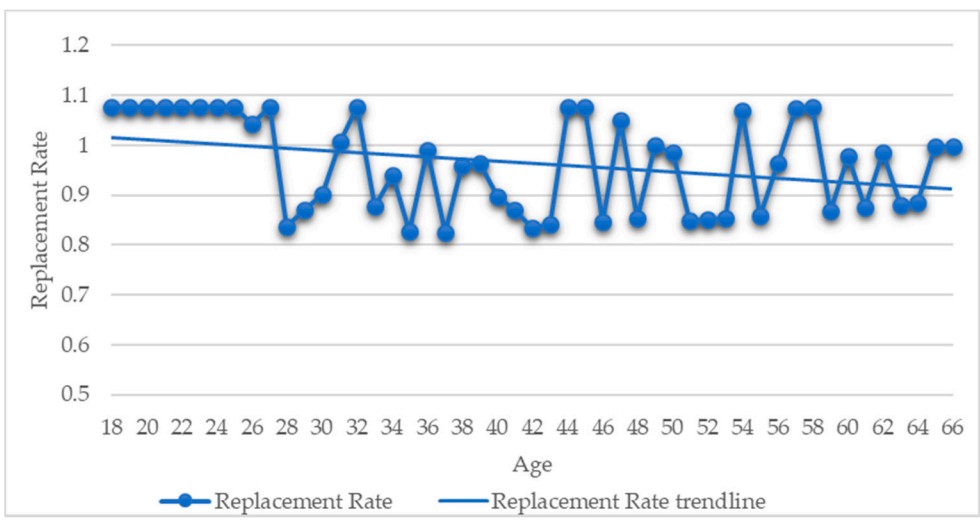

**Figure 5.** Replacement Rate for the unemployed. Own elaboration based on EU-SILC (2016).

For this reason, the subsequent study in relation to poverty has focused on the data provided by the part of the sample corresponding to "workers". In this case, the results obtained show a progressive reduction in the replacement rate, affecting the younger population of the sample to a greater extent. Given that the replacement rate relates the first pension received to the last salary, a decrease in the purchasing power of future pensioners was detected and, therefore, the action of the latest reforms made in the system based on the White Paper on Pensions (2012) of the European Commission [24] is questionable. It provided details of different aspects of the reforms of public pension systems, highlighting the progress of the reforms in achieving sustainability, but showing concern about the benefits they provided. This concern is also reflected in the 2018 Pension Adequacy Report (European Commission, 2018) [25], noting that although measures such as the Sustainability Factor reconcile the sustainability of the system in an ageing population context, they also entail a strong inequality-generating component.

### 4.3. Evolution of Poverty as a Consquence of the Reforms

According to official INE data, in 2016 the poverty rate in Spain was 22.3% and the average income per consumption unit was EUR 15,842. The poverty threshold is not officially published, as an illustration for one-person households INE published EUR 8208.5 that year and for households with two adults and two children was EUR 17,217.9. Both the official rates and the official average incomes do not exactly coincide with the data that we have obtained from the INE's own microdata due to the filtering explained in Section 3. In order to summarize some official microdata and to relate with our results, we calculated the data collected in Table 7.

**Table 7.** Spanish population and poverty data EU-SILC 2016.

| Total Households | Total Persons |
|:---:|:---:|
| 18,408,320 | 45,953,168 |
| Pensioner households (at least one pensioner) | Persons in pensioner households |
| 5,256,310 | 11,335,526 |
| Total households below poverty line | Total persons below poverty line |
| 3,813,549 | 10,268,916 |
| Poverty rate (households) | Poverty rate (persons) |
| 20.7% | 22.3% |
| Pensioner household poverty rate | Persons in pensioner households poverty rate |
| 11.93% | 12.79% |

Source: own elaboration based on EU-SILC microdata.

As can be seen in Table 7, traditionally the poverty rate of households where retirees live is much lower than the national rate. It is evident that having one or more assured incomes significantly reduces the risk of poverty. However, the poverty rate of this subgroup of the population may well be very sensitive to the reforms that were studied in this paper. This being one of the main objectives of this work.

As observed in the previous section, as the gap to retirement increases, the replacement rate and therefore the purchasing power of the future retired population, is increasingly lower. For this reason, we proceeded to study how this decrease in terms of poverty can affect the households in which the retired population is located. To delve into the history, measurement and policies related to poverty, a good source of study can be found in "The Economics of Poverty" [26].

As can be observed in Table 8, some considerations were taken into account for its analysis:

**Table 8.** Methodological issues in the evolution of poverty analysis.

| |
|:---|
| For all the calculations carried out, all cases were weighted by their corresponding factor. |
| In the first place, the poverty threshold was calculated for the initial situation of the sample, defined as 60% of the weighted median of the equivalent income, which is the official criterion applied by the European Union and used in the "Living Conditions Survey Methodology" of INE [27]. As can be seen, it is a concept of relative poverty, different from the absolute concept where the threshold is usually a fixed monetary amount |
| Those households whose equivalent income is above this threshold are considered "not poor" and those whose income is below, are considered "poor" |

Source: own elaboration.

As a result of this first analysis, it was ascertained that the poverty threshold stands at EUR 8299.84, finding that 3,752,077 households, that is, 20.515% of all households nationwide, are below the poverty threshold. If the composition of these households is observed, 635,941 of them (3.5% of the total) have at least one retiree, with 2.25 being the average number of members, and with an average individual income of EUR 6432.14.

Thus, in order to find out the impact of the reforms on household poverty, two hypothetical situations were established in relation to the reduction obtained in the replacement or substitution rate. It should be noted that the results obtained in the previous section on

the average annual pension analysis resulted in future retirement pensions being reduced by between 5% and 8% depending on the time left for retirement. So, at this point, we ask what would happen in relation to household poverty if this reduction were to occur today. To this end, different operations were carried out.

The retirement pension declared by the sample (gross retirement benefit) was reduced by the difference in the replacement rate already mentioned, establishing two alternative scenarios.

- In Scenario 1, the gross retirement benefit was reduced by 5%, and in Scenario 2, the reduction was 8%. Once the retirement benefit to be received was reduced, it was divided by its corresponding consumption unit (this being the methodology followed in studies carried out by the INE and Eurostat). Subsequently, the result was then subtracted from the equivalent household income. Afterwards, for this purpose, it was considered that a household can be made up of more than one pensioner, so, in these cases, the equivalent income was reduced accordingly. Finally, after obtaining the new equivalent household incomes, the new poverty threshold was calculated for each of the scenarios.

As a result, in the case of Scenario 1 (5% reduction), it was obtained that the new poverty threshold stands at EUR 8199.54, so that households whose equivalent income is below this value will be considered "Poor". Thus, in this scenario, it was obtained that 20.467% of households are below the threshold. Among the composition of these households with at least one retired member, 3.7% of the total are below the threshold (that is, 681,448 households with at least one retired member are below the poverty threshold), with an average individual income of EUR 6289.23 and 2.23 people in the household, as can be seen in Figure 6. This significant increase in the number of poor households (45,507) represents an increase of almost one percentage point in the poverty rate.

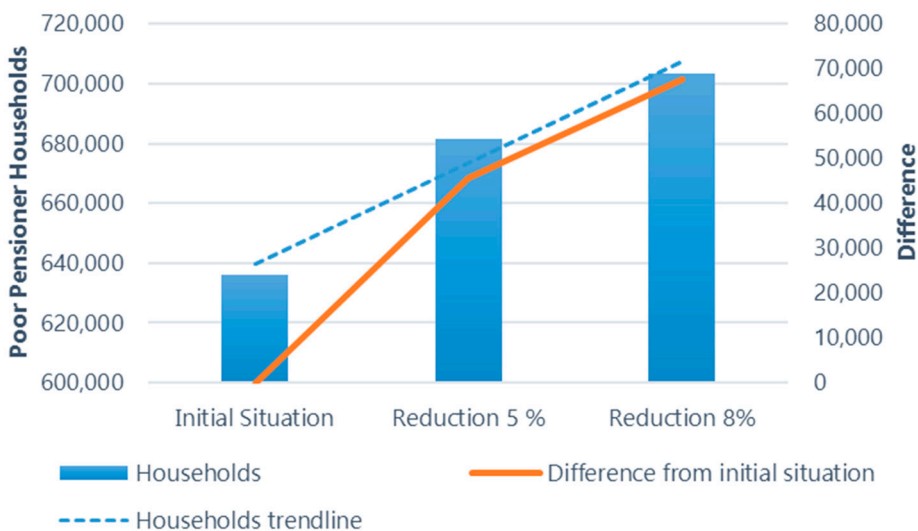

**Figure 6.** Effect of the reforms on the poverty of Spanish households. Own elaboration based on EU-SILC (2016).

Regarding the results obtained for Scenario 2 (8% reduction), the new threshold drops to EUR 8123.29, following the same considerations as in the previous cases, with 20.327% below the poverty threshold. In relation to households made up of at least one retiree, there were 703,475 households, that is, 3.8% of Spanish households, with an average income of EUR 6182.95, and made up of 2.2 people (Figure 6). In this case the increase in poor households (67,534) shows an increase of 1.3 percentage point in the poverty rate.

As explained earlier, since it is a relative concept of poverty, the reduction in the average income (and the median) also reduces the poverty threshold and, therefore, can reduce

the poverty rate. However, the poverty rate that we are interested in observing is the one that affects the group of households with at least one retiree. In this population group, the data show a clear increase in their poverty rate. Figure 6, on left ordinate axis, illustrates the number of poor pensioner households, showing a notable increase in the two scenarios. The right ordinate axis shows the increase in the number of poor pensioner households from the initial situation to each of the scenarios.

## 5. Conclusions

As explained in detail throughout this paper, the concern for the sustainability and adequacy of the Spanish public retirement pension system has been and is very present.

Although the major 2011 and 2013 reforms contribute to improving the system in terms of sustainability, in recent years, concern has grown regarding the adequacy and sufficiency of pensions and their relationship with household poverty. For this reason, this paper includes as its objectives analysis of the effect of the reforms in terms of poverty on the pensions of future retirees, both individually and in the household as a whole.

For this purpose, the EU-SILC 2016 was used, calculating first future retirement pensions by age groups (between 18 and 66), as well as their replacement rate, obtaining as a result a decrease in the replacement rate (relation of the retirement pension with the last salary), finding that, as the moment of effective retirement moves further away, the replacement rate decreases. For an average age group between 41 and 46 years old, the replacement rate is 83%, while for the 60–66 age group which is closest to retirement, the replacement rate is between 87% and 88%. Therefore, when the 41–46 age group reaches retirement, their pensions will show an additional reduction in their purchasing power of 5%. Similarly, for the youngest segment, aged between 23 and 27, this reduction will increase to 8%, obtaining a replacement rate of 80%, so that in their respective moments of access to retirement, their pensions will be reduced to 80% of their last salary.

The official poverty rate according to the INE for 2016 was 22.3% of the population in Spain. In our analysis, the rate obtained is somewhat lower due to the data filtering performed. On the other hand, the poverty rate of the population subgroup of households with some retirees is clearly lower than the national rate. The fact of having an assured income allows fewer households to be below the poverty line. In spite of reaching a lower poverty rate, the affected population is particularly vulnerable. For this reason, in this paper, the sensitivity of the poverty rate in this subgroup was analysed with the intention of studying their response to reforms.

Due to these circumstances, the question of what would happen to Spanish households in relation to relative poverty if this pension reduction had occurred in the income acquisition year (2015) is raised, establishing two alternative scenarios to the initial situation.

The results obtained show that if retirement pensions had been reduced by 5% in 2015, the poverty rate would have hardly varied in general terms, although this variation would be more significant regarding those households with at least one retiree.

This would entail an increase of 45,507 "pensioner" households located below the poverty line. This means an increase of almost one percentage point in the poverty rate of this population group.

Regarding the second scenario, if pensions had been reduced by 8%, the results obtained show a decrease in the general poverty rate. However, in relation to the number of pensioner households located below the poverty line, it increases by 67,534 more households. Thus, in the proposed scenario, the results obtained show a decrease in the general poverty rate, as the income of its components has been reduced, but in relation to the number of pensioner households located below the poverty line, it increases by 67,534 more households compared to "pensioner" households found in the initial situation 1.3 percentage point increase in the poverty rate of this population group).

Therefore, taking into account the results obtained, it can be concluded that the modifications introduced with the 2011 and 2013 reforms of the Spanish pension system have negatively affected the purchasing power of the future retired population, finding a

progressive decrease in the replacement rate and, consequently, increasing the number of "pensioner" households below the poverty threshold.

This paper is focused on the Spanish public pension system in order to analyse the model and the consequences of the reforms on the poverty of the future pensioner population. We are aware of the limitation of our results to the Spanish case. However, this study could be adapted to other neighbouring countries with similar PAYG systems, such as: France, Italy, Portugal, among others, as shown in the paper in the Table 2.

We consider it is a matter of great importance that the different European governments, when seeking the sustainability of their public pension systems, take fully into account the risk of poverty of future pensioners, which could follow an increasing trend in many cases.

**Author Contributions:** Conceptualisation, A.J. and B.R.-C.; methodology, A.J., B.R.-C. and B.U.; software, B.U.; validation, A.J., B.R.-C. and B.U.; formal analysis, A.J., B.R.-C. and B.U.; investigation, A.J., B.R.-C. and B.U.; resources, A.J., B.R.-C. and B.U.; data curation, B.U.; writing—original draft preparation, B.U.; writing—review and editing, A.J., B.R.-C. and B.U.; visualisation, A.J., B.R.-C. and B.U.; supervision, A.J. and B.R.-C. All authors have read and agreed to the published version of the manuscript.

**Funding:** This research was funded by Research Group GR18106 (Public Economics) of University of Extremadura (Spain) and Public Pension System Research Group of University of Extremadura (Spain).

**Institutional Review Board Statement:** Not applicable.

**Informed Consent Statement:** Not applicable.

**Data Availability Statement:** Not applicable.

**Conflicts of Interest:** The authors declare no conflict of interest.

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
