# Peer review of "What Will Retirement Pensions Be Like? Analysis of Spanish Future Pensioner Households in Terms of Poverty"

_sustainability, doi:10.3390/su13041760_

Round 1

Reviewer 1 Report

Referee report about the paper “What will retirement pensions be like? Analysis of Spanish future pensioner households in terms of poverty”, re-submitted for publication in Sustainability.

I thank the authors for their effort to accommodate the manuscript to the reviewers' suggestions. However, some minor suggestions have not completely attended. See my comments below.

Comments

- I think that contents of Tables 1, 3, 4, 5 and 8 should not appear in table form because there is only one column. I suggest to modify it if this change with respect to previous version of the manuscript is not due to an imposition of a present reviewer.

- Replace table 1 by Table 1 in line 99.

- Section 4.2. Figure 2 should be explained better, in a similar way to Figure 4 (lines 242-345). It is not a standard graph. For instance, lines “Age” and “First Pension Limits” don’t appear in the graph. Moreover, there is no difference between colours. Why is there a correspondence between replacement and pension that allows to have two vertical axes simultaneously? They look like two charts in one. Replace “thtat” by “that” in line 242.

It is not necessary to explain the structure of Figure 3 because is as the Figure 2. The graph has changed due to the inclusion of years 18-23. Thus, replace “maintain a continuing trend” by “follow a slightly increasing trend” (line 249). Please, explain briefly why the replacement is not linear unlike the previous case.

- Revise the direct link from the .pdf of reference [24]. It is correct but it is not possible to access the web by clicking on it.

Reviewer 2 Report

The authors' responses to the comments were sufficient. I believe the manuscript improved and I recommended this paper to be accepted.

Author Response

Thank you very much for your comments and suggestions. The work improved a lot.

This manuscript is a resubmission of an earlier submission. The following is a list of the peer review reports and author responses from that submission.

Round 1

Reviewer 1 Report

Comments and Suggestions for Authors

The paper covers important issues like the long-term sustainability of pension systems, pension reforms and their consequences on the financial well-being of retirees. However, I think some more information and discussion are necessary to interpret the findings. Here are a few comments.
Most countries have witnessed a shift from defined benefit to defined contribution pension systems over the last decades. It may be interesting to know whether this change has been discussed in Spain as well.

The authors report that two major reforms carried out in 2011 and 2013 increased the legal retirement age to 67. However, in many countries it is possible to retire before reaching such an age. Which are the ways to access early retirement in Spain? It may be a widespread option, depending on its eligibility requirements. Details on the Spanish pension system are helpful in order to better understand the context in which the reforms were introduced.
The age of entry in the labor market is considered to be 18. How was this threshold chosen? The age of 18 is unlikely for more educated individuals. Clearly the authors look at averages, but taking into account the educational dimension may be interesting in future work. Also, could you be more precise on how you projected individuals’ last salary before retirement?

The authors define the poverty threshold as 60% of the weighted median of the equivalent income. However, this definition is given just before the conclusions. I suggest that the authors mention it at the beginning of the paper to make the reader aware of the fact they are referring to relative poverty.

To conduct their analysis on poverty, the authors exploit the gross retirement benefit declared by the individuals interviewed in the EU-SILC 2016 for Spain. Since the retirement pension is self-declared, I think the authors should acknowledge that it is not obvious that individuals know their future public pension.

The authors find that the pension reforms decrease the replacement rates. In order to contextualize the results, I think it would be useful to mention the current replacement rates in some other comparable countries. In fact, Spain has one of the highest gross pension replacement rates among the OECD countries (OECD, 2019).

All the references are in Spanish. It would be helpful to include works published also in English, if any.

On a minor note, a few typos were found in the text, e.g., at lines 44, 45, 62, 175, 191, and 379.

Reference:
OECD (2019), Pensions at a Glance 2019: OECD and G20 Indicators, OECD Publishing, Paris.

Reviewer 2 Report

The present version is totally centered in the Spanish system. Are there some implications for other countries? Otherwise the audience is limited. In addition to Spain should be in the title, because this is not a general paper.

Some of the last sentences in the conclusions are not connected to the results. Please check and make the message go to the point.

Minor point:

There is an excessive number of bullet points. Can you put information in Tables if you want to present lists? Otherwise, avoid excess of item lists.

Compare the results on income and poverty to regular information sources from the Spanish National Institute of Statistics.

Reviewer 3 Report

Referee report about the paper “What will retirement pensions be like? Analysis of future pensioner households in terms of poverty”, submitted for publication in Sustainability.

This paper analyzes the current situation of the Spanish public retirement pension system and its effect on the future retired population. In order to get the long-term viability, that is to say, the Spanish Government realized two reforms of the system. The pay-as-you-go system and the correlation between contributions and benefit are maintained. The effect of the two reforms of the system on the future pensioners has been analyzed by looking at the evolution by age of the average annual pension and the replacement rate and the relative power by age groups.

The analysis of the viability of a public pension system is interesting in finance and for a potential reader of Sustainability. The authors analyze in detail the recent references used and explain the calculations from real data. The paper is well organized. I think that the paper can be interesting for the journal.

Comments

- Abstract. It is some long, I suggest lighten by excluding the methodology and the conclusions.

- I don’t know of the counter reforms 2018-2020 have been taking into account in the analysis and result section.

- Section 4.1. The average annual income decreases along 60-66. It must be explained and indicated if it produces an effect on pensions. Replace Graph 1 by Figure 1 in line 213.

- Section 4.2. Replace Graph 2 by Figure 2 in line 250. Eliminate the first (2016) in line 253. Figure 2 should be explained better. For instance, lines “Age” and “First Pension Limits” don’t appear in the graph. Why is there a correspondence between replacement and pension? They look like two charts in one. Ages 18-23 have been excluded, and then some results are included in Table 1, but it is not cited in the text. Replace 88 by 89, in line 260, and 80 by 81, in line 262. Replace Graph 3 by Figure 3 in lines 272, and thtat by that in line 273. I think that the trend is constant (line 279). Explain Figure 3.

 - A comment about a comparison with other countries based on pay-as-you-go and defined benefit can improve the paper (the replacement rates, the average annual pensions, the relative power and the reforms and conditions of their pension systems).

- Section 4.3. Figure 4 is not cited and is not explained.

- Conclusions. Some comments on the results of Section 4 are repetitive.

Round 2

Reviewer 2 Report

The main points of my review have no been addressed. (i) The content of the paper is still too restricted to the Spanish case. Limited interest to international readers. (ii) The revision of the conclusions is insufficient. (iii) The style is still too much like a report with an excessive number of bullet points.  (iv) even if some official figures have been introduced, there are still not at all connected to the results.